# Cardioprotective Effects of 6-Gingerol against Alcohol-Induced ROS-Mediated Tissue Injury and Apoptosis in Rats

**DOI:** 10.3390/molecules27238606

**Published:** 2022-12-06

**Authors:** Venkata Subbaiah Ganjikunta, Ramana Reddy Maddula, Shanmugam Bhasha, Ravi Sahukari, Shanmugam Kondeti Ramudu, Venkatrayulu Chenji, Sathyavelu Reddy Kesireddy, Zhe Zheng, Mallikarjuna Korivi

**Affiliations:** 1Division of Molecular Biology and Ethanopharmacology, Sri Venkateswara University, Tirupati 517 502, India; 2Department of Zoology, PRR & VS Government Degree College, Vidavalur 524318, India; 3Department of Marine Biology, Vikarama Simhapuri University, Nellore 524320, India; 4Exercise and Metabolism Research Center, College of Physical Education and Health Sciences, Zhejiang Normal University, Jinhua 321004, China

**Keywords:** apoptosis, ROS, oxidative stress, protein carbonyl, gingerol, ethanol

## Abstract

The present study investigated the cardioprotective properties of 6-gingerol against alcohol-induced ROS-mediated cardiac tissue damage in rats. Experiments were conducted on 4 groups of rats, orally treated with control, 6-gingerol (10 mg/kg body weight), alcohol (6 g/kg body weight) and combination of 6-gingerol plus alcohol for two-month. In the results, we found 6-ginger treatment to alcohol-fed rats substantially suppressed ROS production in cardiac tissue. Alcohol-induced elevated 8-OHDG and protein carbonyls which represent oxidative modification of DNA and proteins were completely reversed by 6-gingerol. This was further endorsed by restored superoxide dismutase and catalase activities with 6-gingerol against alcohol-induced loss. The elevated cardiac biomarkers (CK-MB, cTn-T, cTn-I) and dyslipidemia in alcohol-intoxicated rats was significantly reversed by 6-gingerol. Furthermore, alcohol-induced apoptosis characterized by overexpression of cytochrome C, caspase-8 and caspase-9 was diminished with 6-gingerol treatment. Transmission electron microscope images conferred the cardioprotective properties of 6-gingerol as we have seen less structural derangements in mitochondria and reappearance of myofilaments. Our findings conclude that 6-ginger effectively protect alcohol-induced ROS-mediated cardiac tissue damage, which may be due to its potent antioxidant efficacy. Therefore, 6-gingerol could be a potential therapeutic molecule that can be used in the treatment of alcohol-induced myocardial injury.

## 1. Introduction

Chronic alcohol consumption is implicated in risk of developing various diseases, including cardiovascular diseases (CVD), and leads to alcohol-related morbidity and mortality [1]. Alcohol and its metabolites can cause mild to severe tissue injuries, mainly due to impaired antioxidant status and oxidative stress [2,3]. The highly unstable free radicals or reactive oxygen species (ROS) that are produced during alcohol metabolism ruins the defensive antioxidant status, and cause destructive damage to cardiac tissue [4]. Excessive accumulation of ROS disrupts the redox homeostasis and leads to the oxidative modification of vital biomolecules, including proteins, lipids and DNA [5,6,7]. Oxidative stress is considered to be an important factor to promote cell death in response to a variety of signals and pathophysiological situations. Excessive alcohol consumption is linked with abrupt apoptosis of cardiomyocytes via various signaling pathways [8,9]. A study found that alcohol induced mitochondrial apoptosis in cardiomyocytes through stimulated oxidative stress [5,10]. Alcohol-induced cardiac tissue damage was further witnessed by elevated cardiac biomarkers, such as creatine kinase-MB (CK-MB), cardiac troponin-T (cTn-T) and cardiac troponin-I (cTn-I) in rats [11]. Although ROS scavengers have been shown to inhibit alcohol-induced cardiac tissue damage [11], the mechanism and involved molecules behind this phenomenon remain elusive.

The rhizome of *Zingiber officinale,* known as ‘ginger’, has been consumed worldwide as a spice and herbal medicine, and cultivated in most tropical regions of the world. Ginger has vital non-volatile pungent phytochemicals, such as gingerols, shogaols, paradols and zingerone, which have shown biological activities [12]. Among them, 6-gingerol (Figure 1A), a phenol phytochemical constituent of fresh ginger, has been reported to possess several pharmacological efficacies, including anticancer, antioxidant, anti-inflammation, anti-platelet aggregation and antifungal [11,13,14]. Sampath et al. reported that 6-gingerol prevents atherosclerosis by inhibiting the oxidative stress biomarkers [15]. Another study showed that 6-gingerol treatment improves antioxidant status and prevents myocardial injury induced by doxorubicin [16]. The protective effects of 6-gingerol against myocardial fibrosis possibly associated with increased antioxidant capacity, decreased inflammation and apoptosis in mice [17].

Although previous studies have revealed the pharmacological efficacies of 6-gingerol against oxidative stress and inflammation, the function of 6-gingerol against alcohol-induced myocardial injury and oxidative stress has not yet been established. Therefore, in this study, we investigated the cardioprotective properties of 6-gingerol against alcohol-induced myocardial damage in rats. We examined the changes in key biomarkers that are involved in regulation of antioxidant status, oxidative stress and apoptosis with or without 6-gingerol treatment in alcohol-fed rats.

## 2. Results

### 2.1. Six-Gingerol Suppress ROS Production in Alcohol-Fed Rats

ROS, which play a crucial role in provocation of oxidative stress, were determined in cardiac tissues of all experimental groups by dihydroethidium (DHE) fluorescent dye. We found ROS distribution in a small area of cardiac tissue in control and 6-gingerol-treated rats (Figure 1B). By contrast, ROS were found in large areas of cardiac tissue in alcohol-treated rats, detected by red fluorescent dye (DHE) under the fluorescent microscope. However, 6-gingerol treatment in combination with alcohol substantially suppressed the ROS generation in cardiac tissue (Figure 1B). These findings reveal that occurrence of ROS-mediated myocardial damage in alcohol-fed rats could be attenuated by 6-gingerol treatment.

### 2.2. Treatment of 6-Gingerol Restores Antioxidant Enzymes against Alcohol-Induced Loss

The concentrations of intracellular ROS typically dictate the tissue antioxidant status. Therefore, we examined the changes of primary antioxidant enzyme activities in response to alcohol and gingerol treatments. In the results, we found a significant decrease of both superoxide dismutase (SOD) and catalase (CAT) activities in cardiac tissue of alcohol-fed rats (Figure 2A,B). Nevertheless, gingerol treatment considerably restored the SOD (*p* < 0.05) and CAT (*p* < 0.05) activities against alcohol-induced loss. The restored antioxidant enzyme activities with gingerol treatment were further witnessed in the inhibition of excessive ROS production.

### 2.3. Gingerol Treatment Reverses Elevated 8-Hydroxyguanosine (8-OHDG) and Protein Carbonyls

We next determined the levels of oxidative damage to DNA and proteins in alcohol- and gingerol-treated groups. Our results showed a considerable DNA oxidative damage in cardiac tissue by alcohol intoxication, which was represented by elevated 8-OHDG levels. This was about 3-fold higher compared with normal control (Figure 3A). It is worth noting that gingerol treatment substantially (*p* < 0.05) inhibited the elevated 8-OHDG levels, indicating the protective effects of gingerol against DNA damage.

Furthermore, we found that alcohol intoxication led to oxidative modification of cardiac proteins, which was evidenced by increased protein carbonyl levels. However, gingerol treatment to alcohol-fed rats significantly (*p* < 0.05) suppressed protein carbonyl content in the myocardial tissue (Figure 3B). Taken together, gingerol is able to protect the cardiac tissue by inhibiting the oxidative modification of DNA and proteins against alcohol intoxication.

### 2.4. Antilipidemic Properties of 6-Gingerol in Alcohol-Fed Rats

To elucidate the antilipidemic effects of gingerol, we measured the changes of total cholesterol (TC), triglyceride (TG), low-density lipoprotein (LDL) and high-density lipoprotein (LDL) levels in all experimental groups. We found that circulating total cholesterol, TG and LDL were significantly increased, while HDL levels were significantly decreased following alcohol drinking. In contrast to these, gingerol treatment to alcohol-fed rats significantly decreased the total cholesterol, TG and LDL levels and restored the HDL levels (Table 1).

### 2.5. Gingerol Prevents Alcohol-Induced Elevation of Cardiac Biomarkers

The protective effects of gingerol against alcohol-induced cardiac tissue injury was determined by measuring the important cardiac biomarkers, including CK-MB, cTn-T and cTn-I. As we assumed, alcohol intoxication significantly elevated the circulating levels of CK-MB (*p* < 0.05), cTn-T (*p* < 0.05) and cTn-I (*p* < 0.05), which indicates the cardiac tissue damage (Figure 4A,B,C). Six-gingerol treatment prevents the alcohol-induced cardiac damage through suppressed cardiac biomarkers (*p* < 0.05). The suppressed levels of cTn-T, cTn-I and CK-MB with gingerol treatment were 41.5%, 40.26% and 42.38%, respectively, compared to alcohol-induced elevation (Figure 4). These results revealed the potential cardioprotective effects of 6-gingerol against alcohol intoxication.

### 2.6. Gingerol Treatment Attenuates Upregulated Apoptotic Markers in Alcohol-Fed Cardiac Tissue

We extended our studies to evaluate the anti-apoptotic properties of 6-gingerol against alcohol-induced induction of apoptosis in cardiac tissue. The studied key apoptotic makers through immunohistochemistry visualized that alcohol drinking triggered the expression of cytochrome C, caspase-8 and caspase-9 in cardiac tissue. Interestingly, 6-gingerole treatment for 60 days along with alcohol diminished the overexpression of cytochrome C, caspase-8 and caspase-9 in cardiac tissue. These findings evidenced that alcohol-induced initiation of apoptosis could be attenuated by 6-ginferol treatment (Figure 5). Similar to antioxidant enzyme activities, the immunohistochemistry images also showed restored expression of SOD and CAT with 6-gingerol co-administration with alcohol (Figure 6A,B).

### 2.7. Transmission Electron Microscopy (TEM) Demonstrates Cardioprotective Effects of Gingerol

Subsequently, TEM was performed to assess the cardioprotective efficacies of 6-gingerole. As shown in the images, control heart was characterized by typical symmetric myofibrils comprised of Z lines with sarcomeres. Furthermore, packed mitochondria adjacent to the fibers were evidenced in the control group. Gingerol alone treated rats shown mitochondria slightly deviated from the fibers remaining myofibril structure intact. However, alcohol alone treated group represented with altered myofibril structure, disappearance of Z lines, derangement of sarcomeres and mitochondrial swelling. All these destructive features imply the mitochondrial apoptosis with alcohol intoxication. It should be noted that gingerol treatment prominently attenuated these structural derangements. These protective effects were evidenced by less structural alterations in mitochondria, reappearance of myofilaments and Z lines (Figure 7).

## 3. Discussion

In this study, for the first time we have demonstrated that 6-gingerol substantially alleviated the alcohol-induced oxidative damage, myocardial injury and apoptosis. Experimental evidence shows that 6-gingerol treatment suppressed alcohol-induced ROS production in cardiac tissue. This therapeutic effect was accompanied by restored myocardial antioxidant status, and decreased DNA oxidative damage and protein carbonyls. In addition, alcohol-induced elevated cardiac biomarkers (CK-MB, cTn-T, cTn-I) and dyslipidemia were significantly reversed by gingerol. Histopathological evidence further confirmed the tissue protective effects of 6-gingerol against alcohol-induced myocardial injury and apoptosis. Since alcohol is typically involved in impairing of antioxidant homeostasis and myocardial injuries, supplementation of antioxidant rich molecules like 6-gingerol could possibly alleviate the myocardial damage. The calculated 6-gingerol dose from our rat model (10 mg/kg) is ~97 mg per adult with an average weight of 60 kg [18]. This dose might be feasible to humans, as a previous study reported no serious adverse effects of 6-gingerol with dose ranges from 100 mg to 2000 mg per adult [19].

It has been documented that alcohol intake, either chronic or acute, alters the cardiomyocyte and whole organ functioning of the heart [20]. ROS or free radicals that are generated during alcohol metabolism are crucially involved in the onset of myocardial disturbances possibly through ruining of intracellular antioxidant homeostasis [21,22]. Accumulations of highly reactive ROS destabilize the myocardial contractility, cause myocardial tissue injury, and induce myocyte apoptosis [23]. In agreement with these findings, we also noticed excessive production of ROS and subsequent myocardial injury in alcohol-fed rats. However, 6-gingerol treatment to alcohol-fed rats for a period of 2 months considerably inhibited the ROS production in cardiac tissue. Previous studies have shown that 6-gingerol is able to decrease ROS-mediated oxidative stress in the liver [24], and its antioxidant property reduces acute renal toxicity [25]. Our previous in vitro findings also demonstrated the free radical scavenging ability of 6-gingerol [26]. A recent study reported that only 7 days of 6-gingerol treatment to arsenic trioxide injected mice significantly decreased ROS production in heart [27]. These findings indicate that 6-gingerole is able to inhibit intracellular ROS production during ethanol metabolism.

Oxidative stress is an imbalance between the amount of intracellular ROS and antioxidant defense status. Alcohol intoxication is characterized by elevated oxidative stress and decreased antioxidant enzyme status in various tissues, including liver and heart of rodents [6,11,28]. Here, we found alcohol-induced oxidative damage of DNA and proteins as reported by elevated 8-OHDG and protein carbonyl levels, respectively. Chronic ethanol consumption in rats has been shown to increase protein oxidation in parallel with increased free radicals [29]. The uncontrolled-highly unstable ROS that are generated during alcohol metabolism eventually affect the function of cardiomyocytes/heart through irreversible damage of mitochondria, DNA and proteins. These events are clinically associated with pathogenesis of cardiomyopathy [5,30]. Therefore, controlling alcohol-induced oxidative stress by known antioxidants is necessary to prevent the progression of cardiomyopathy. We found that gingerol treatment along with alcohol tremendously inhibited the formation of 8-OHDG and protein carbonyls. Due to their highly reactive nature, hydroxyl radicals or singlet oxygen hydroxylate the deoxyguanosine and formed 8-hydroxy-2′deoxyguanosine. Decreased 8-OHDG levels with 6-gingerol explains that gingerol molecules possibly inhibit the production of intracellular hydroxyl radical and/or singlet oxygen against alcohol toxicity and thereby prevent the oxidative damage to DNA and proteins. Our present findings and previous reports also confirmed the potent free radical scavenging activity or ROS inhibitory efficacy of 6-gingerol [26]. An in vitro study reported that 6-gingerol possess a strong protective ability against the DNA damage caused by mono (2-ethylhexyl) phthalate in human endothelial cells, and the mechanism may be due to its antioxidant activity [31].

Decreased intracellular ROS production or oxidative stress in any context could possibly explained by the antioxidant status of a tissue. Therefore, we determined the primary antioxidant enzyme activities in the heart of alcohol and 6-gingerol-treated rats. In accordance with elevated oxidative stress and ROS production, the primary radical scavenging enzymes SOD and CAT were significantly decreased following alcohol intoxication. Decreased antioxidant enzyme activities with alcohol drinking were repeatedly confirmed by several previous studies [3,5]. Persistent lower grade antioxidant capacity and subsequent oxidative stress perhaps contributes to morphological changes and malfunctioning of cardiac tissue [5]. The important finding in our study is that the decreased SOD and CAT activities were restored with 6-gingerol treatment. In a rat model, 6-gingerol was reported to increase antioxidant enzyme activities and protect the intestinal barrier from ischemia/reperfusion-induced damage [32]. Various doses of gingerol fraction was reported to improve the renal antioxidant enzyme activities and decrease the lipid peroxidation against gentamicin-induced nephrotoxicity in rats [25]. Another study demonstrated that cotreatment of 6-gingerol and ethanol significantly restored the antioxidant enzyme activities against ethanol-induced loss in cultured mouse embryos [33]. Taken together, our findings provided additional knowledge that 6-gingerol treatment for a period of 2 months could protect the cardiac tissue against alcohol-induced oxidative damage through improved antioxidant status.

Chronic alcohol consumption not only impairs the redox homeostasis, but also causes dyslipidemia, a major risk factor for developing cardiovascular diseases. Dyslipidemia is typically represented by abnormal increase of circulating lipid profiles; including TG, TC and LDL, while HDL levels are decreased. Alcohol administration to rats resulted in increased TG, TC and LDL levels and decreased HDL, which confirmed the dyslipidemia status. In our study, 6-gingerol treatment to alcohol-fed rats reversed the dyslipidemia as shown by decreased TG, LDL, and TC levels along with increased HDL levels. Clinical studies have shown that increasing TG, LDL and TC increases the risk of non-ischemic heart failure, while decreasing serum lipids can reverse the heart dysfunction [34]. In this context, decreased levels of TG, LDL and TC with 6-gingerol treatment perhaps improved the heart function which was deteriorated by alcohol intoxication in rats. Saravanan and colleagues also reported that oral administration of gingerol for 30 days contributed to a significant decrease of lipid profile in obese rats [35]. Another study reported that 6-gingerol treatment for 12 days significantly decreased plasma TG, LDL, and free fatty acid concentrations in db/db mice. This lipid-lowering effect of 6-gingerol was accompanied by an increased antioxidant enzyme activities and suppressed ROS generation in db/db mice [36]. These findings reveal that 6-gingerol is a considerable molecule to lower the lipid profile and prevent the lipid-associated cardiac dysfunction.

Circulating cardiac biomarkers are novel quantitative measures which shed light on occurrence of cardiac pathophysiology [37]. The biomarker troponin I (cTn-I/cTn-T) is a sensitive and specific marker for myocardial injury, which is widely used to diagnose the myocardial infarction [38]. In our study, alcohol-induced elevation of clinical cardiac biomarkers in plasma is another evidence of cardiac tissue damage in alcohol-fed rats. There was a substantial correlation between elevated lipid profile and increased release of cardiac biomarkers in plasma by damaged cardiac cell under ethanol intoxication [11]. In agreement with previous findings, our results also revealed that ethanol intoxication increase the cardiac biomarkers, including CK-MB, cTn-T and cTn-I in plasma. A previous study reported a several-fold rise in serum troponin, myoglobin and CK-MB concentrations after alcohol septal ablation for the treatment of hypertrophic cardiomyopathy [39]. Based on the cardiac biomarker data, a community-based cohort study reported even modest habitual alcohol intake was associated with an increased risk of atrial fibrillation [37]. These results indicate that controlling of cardiac biomarkers is essential in treating the myocardial injuries. Nevertheless, 6-gingerol treatment along with alcohol exerts the cardioprotective effects by diminishing the elevated CK-MB, cTn-T and cTn-I levels. A study by Ren et al., demonstrated that various concentrations of 6-gingerol protects cardiomyocytes from hypoxia-induced injury by suppressing the release of lactate dehydrogenase (LDH) and apoptosis [40]. Cotreatment of ginger to alcohol-fed rats also reported decreased concentrations of cTn-T, LDH and CK-MB in rats [11]. Pretreatment of 6-gingerol has been shown to improve the cardiac function by decreasing the cTnT and CK-MB expressions in ischemia/reperfusion (I/R)-induced myocardial injury model [41]. Lipid-lowering effect and/or improved antioxidant status by gingerol possibly contributed to attenuate the alcohol-induced elevation of cardiac biomarkers.

It has been speculated that overwhelming production of ROS intrinsically involved in initiation and progression of cardiac myocytes apoptosis under chronic ethanol intoxication [42]. Several studies have shown that chronic alcohol consumption promptly associated with occurrence of cardiomyocytes apoptosis by alterations in mitochondria membrane potential [43,44]. Although we haven’t detected the mitochondrial membrane potential, the apoptotic proteins in implicated in mitochondrial apoptosis was determined through immunohistochemistry. Our results showed that the apoptotic biomarkers, such as cytochrome C, caspase-9 and caspase-8, were highly expressed in alcohol-intoxicated hearts. Furthermore, elevation of apoptotic proteins were accompanied by decreased cardiac SOD and CAT expressions with alcohol. At molecular levels, ethanol may initiate the apoptosis in cardiac tissue by both intrinsic and extrinsic pathways. In the cascades of apoptosis, ethanol-induced ROS accumulation and decreased antioxidant defense mechanism may play a major role in the expression of caspase-8. Triggering of cytochrome C release by alteration in mitochondrial membrane permeability led to activation of caspase-9 [45,46]. Meanwhile, 6-gingerol treatment profoundly inhibited the apoptosis in cardiac tissue by suppressing the overexpressed cytochrome C, caspase-9 and caspase-8, while restoring the tissue antioxidant status. The earlier reports have shown that ROS scavengers or inhibitors can control the alcohol-induced apoptosis in cardio myocytes [47]. In this context, 6-gingerol was reported to decrease the formation of apoptotic proteins that reserved the myocardial apoptosis under hypoxia condition [40]. 6-gingerol administration to isoproterenol-treated mice (14-day) resulted in a significant reduction of caspase-3, Bax protein expressions and Bax/Bcl-2 ratio, indicating attenuation of cardiac apoptosis. This anti-apoptotic property of gingerol was accompanied by inhibition of oxidative stress and inflammatory biomarkers [17]. It has been reported that 6-gingerol metabolites may stay about 12 h in the body after oral administration [48]; therefore, we assume that the bioavailability of gingerol may be responsible for its therapeutic efficacies under stress. Taken together, our results suggest that 6-gingerol can inhibit the apoptosis under ethanol intoxication through reduction of ROS production and improvement of antioxidant defense mechanism.

Another important finding of this study is that TEM analyses provided additional convincing evidences that 6-gingerol is a potent tissue protective molecule. TEM visuals demonstrated that ultra-structural derangement in cardiac tissue following ethanol intoxication. We further noticed intense lysis of myofibrils with disruption of Z lines and sarcomere structure, mitochondrial swelling and damage, which collectively results in myocardium apoptosis in ethanol-intoxicated rats. It is worth noting that 6-gingerol attenuate all these destructive changes and retained the ultra-structure of myocardium against cardiotoxic effect of ethanol. In I/R-induced myocardial injury model, 6-gingerol pretreatment to rats reduce myocardial infarction area and degree of cardiac pathological injury followed by lower levels of myocardial enzyme and inflammatory mediators [41]. Some studies have shown that food-based formulations that contain ginger could promote alcohol metabolism and alleviate the alcohol hangover effects [49]. In such case, we assume that 6-gingerol may be involved in rapid alcohol metabolism, which facilitates lower accumulation of toxic alcohol metabolites and then results in less cardiac tissue damage. Despite the lack of experimental evidence to reveal this phenomenon, this is the first report to demonstrate the cardioprotective effects of 6-gingerol against ethanol-induced cardiac tissue damage in rats.

## 4. Materials and Methods

### 4.1. Chemicals and Reagents

The phytochemical compound, 6-gingerol (purity: >98.0%) was purchased from TCI chemicals (India) Pvt. Ltd. Both dihydroethidium fluorescent dye and SOD primary antibody were obtained from the Sigma-Aldrich (St. Louis, MO, USA). Catalase, caspase-9 and -8, cytochrome C primary antibodies and secondary antibody horseradish peroxidase (HRP)-conjugated goat anti-rabbit IgG were purchased from Invitrogen (Waltham, Thermo Scientific, MA, USA). Troponin-T primary antibody was obtained from Abcam (Cambridge, UK). Creatine kinase–MB (CK-MB), cardiac troponin-T (cTn-T) and cardiac troponin-I (cTn-I) ELISA kits were purchased from the Biocheck, South San Francisco, USA. Triglyceride (TGs), total cholesterol (TC), high-density lipoprotein (HDL) and low-density lipoprotein (LDL) colorimetric kits were obtained from the Merck (Rathway, NJ, USA). All other chemicals and reagents were analytical reagent grade and purchased from Sigma-Aldrich.

### 4.2. Experimental Animals

A total of 24 Wister strain male albino rats, weighing 200 ± 30 g, were obtained from the Indian Institute of Science, Bangalore, India. All rats were housed in standard polypropylene transparent cages at a controlled temperature (23–27 °C) and humidity (50% ± 5%) with a 12 h light/dark cycle. Rats had free access to food and water. The experimental design and protocols were approved by the Institutional Animal Ethics Committee (IAEC) of Sri Venkateswara University, India, with (10/i/a/CPCSEA/IAEC/SVU/KSR- GVS/dt 15 November 2010).

### 4.3. Animal Grouping and Treatment

After a week of adaptation to the laboratory conditions, the rats were randomly assigned into 4 groups, control (C), 6-gingerol (Gt), ethanol (Et) and combination of both ethanol plus 6-gingerol (Et + Gt) groups, with 6 rats in each. Rats in the control group received 0.9% saline solution, rats in the Gt group received 6-gingerol by gavage (10 mg/kg bodyweight, oral administration) and rats in the Et group received ethanol by gavage (20%, 6 g/kg bodyweight, oral administration). The remaining rats in the Et + Gt group received both ethanol and 6-gingerol as described in respective groups. The treatment was given once a day for all groups, and the whole treatment lasted for 2 months. The selective dosage of alcohol and 6-gingerol used in this study was well established by previous studies [11,16]. Blood samples were collected at 10 am into a sterile heparinized tube from the retro-orbital plexus of rats at the end of the treatment period, and plasma was collected by centrifugation at 2000 rpm for 10 min at 4 °C and then stored at −20 °C for further analysis. The heart tissues were isolated from rats after cervical dislocation and immediately washed with ice cold saline and kept on a chilled glass plate. One part of the heart tissue was stored at −20 °C for further biochemical assays, and the remaining was fixed in 4% formaldehyde for histopathological studies.

### 4.4. ROS Detection by Dihydroethidium (DHE) Fluorescent Dye

ROS in the cardiac tissue was detected using DHE dye as described by Carter et al., (1994) method [50]. Cryostat sections of cardiac tissue was cut into 10 µm thick and incubated with 5M DHE dye at 37 °C for 30 min. The DHE is oxidized by ROS to produce fluorescent ethidium that subsequently binds to nucleic acids, further staining the nucleus a bright fluorescent red. The red fluorescent from DHE was detected by fluorescent microscope.

### 4.5. Determination of 8-Hydroxydeoxyguanosine (8-OHDG)

Wash solution (20×) in a volume of 15 mL was diluted with 285 mL of distilled water and prepared 1x assay buffer by 14 mL of assay buffer was added to 56 mL of distilled water. To each well, cardiac tissue samples or standard 50 µL and 75 µL of conjugate was added. With the exception of NSB wells, 25 µL of 8-hydroxydeoxyguanosine antibody was added, and then the side of the plate was tapped to mix and incubated for 2 h at room temperature and 100 µL of TMB substrate was added to each well. The substrate solution began to turn blue, was incubated for 30 min at room temperature without shaking, 50 µL of stop solution was added to each well and the absorbance was read at 450 nm. The plate reading was completed within 10 min after adding the stop solution.

### 4.6. Estimation of Protein Carbonyl Content

Protein carbonyl content in the cardiac tissue was estimated by commercial kit (Sigma-Aldrich). Briefly, 200 mg of tissue was homogenized in 1 mL of tris buffer and centrifuge at 10,000× *g* for 15 min at 4 °C, and the supernatant was stored on ice. A 200 µL sample was added to 800 µL of DNPH, incubated for 1 h in dark at room temperature, and each tube was vortexed briefly for every 15 min during incubation, 1 mL of 20% TCA was added to each tube, and the tubes were vortexed, placed on ice, incubated for 15 min and centrifuged at 10,000× *g* for 10 mint at 4 °C. The supernatant was discarded, the pellet was re-suspended in 1 mL of TCA and the tubes were placed on ice for 5 min and centrifuged at 10,000× *g* for 10 min. The supernatant was discarded, and the pellet was resuspended in 1 mL of (1:1) ethanol and ethyl acetate mixture, then centrifuged at 10,000× *g* for 10 min at 4 °C (Remi C-24 BL/CPR 24 Cooling centrifuge, Mumbai, India). A 220 µL sample was added to each well, and the absorbance was measured at 360–385 nm (Simadzu, UV1800 Spectrophotometer, Kyoto, Japan).

### 4.7. Assessment of Superoxide Dismutase and Catalase Activities

Changes in SOD activity in the mitochondrial fraction was assessed by the method of Misra and Fridovich (1972) [51] at 480 nm for 4 min on a Simadzu, UV1800 Spectrophotometer (Kyoto, Japan). Activity was expressed as the amount of enzyme that inhibits the oxidation of epinephrine by 50%, which is equal to 1 U per milligram of protein. We then determined the CAT activity at room temperature using the modified version of Aebi [52]. The absorbance of the sample was measured at 240 nm for 1 min in a UV-Spectrophotometer (Simadzu, UV-Spectrophotometer, Kyoto, Japan). One-unit activity is equal to the moles of H_2_O_2_ degraded/mg protein/min.

### 4.8. Estimation Cardiovascular Risk Factors

Widely considered lipid based cardiovascular risk factors, including triglyceride (TGs), total cholesterol (TC), high density lipoprotein (HDL) and low-density lipoprotein (LDL) levels, were measured in plasma by commercially available kits (Merck, Rathway, NJ, USA).

### 4.9. Evaluation of Cardiac Biomarkers

The isoenzyme creatine kinase–MB (CK-MB) levels in plasma were estimated using a commercially available kit (Biocheck, ELISA kit, South San Francisco, CA, USA) in all experimental groups. The absorbance of the sample was measured at 340 nm using iMark Microplate reader (Bio-Rad, Hercules, CA, USA). CK-MB levels were described as international units per liter. Next, changes in cardiac troponin–T (cTn-T) and cardiac troponin–I (cTn-I) in plasma were determined by commercially available kits (Biocheck ELISA kit, South San Francisco, CA, USA) in all experimental groups. The sample absorbance was measured at 450 nm in an iMark Microplate absorbance reader (Bio-Rad, Hercules, CA, USA). The units of cTn-T and cTn-I levels were described as grams per milliliter.

### 4.10. Immunohistochemistry

Immunohistochemistry study was performed according to the study by Liu et.al., [53]. Briefly, the sections were collected on 3-aminopropyltriethoxysilane (APES)-coated slides, and xylene was added and kept for 15 min. For each slide, alcohol was added in descending concentrations (100–70%), and after 1 min, the slides were washed in tap water for 10 min and distilled water for 5 min. Further, the slides were kept in a presser cooker in 1 mM of citrate buffer (pH 6). After leaving the pressure cooker, placed in sink water for cooling at room temperature, the samples were washed in distilled water for 5 min and with 5 mM of TBS buffer (pH 7.6) twice for 5 min each time. The samples were blocked with peroxidase for 10–15 min, washed three times with TBS buffer and drained. The section was covered with concerned primary antibodies (caspase-8, caspase-9 and cytochrome C) for 1 h (1:1000). Further, the samples were washed three times with TBS buffer for 5 min, andthen incubated with the HRP-conjugated secondary antibody (1:1000). The color was developed within 5–8 min with the use of 3,3′-diaminobenzidine. The samples were washed with TBS buffer three times for 5 min, then incubated in hematoxylin for 1 min. Finally, samples were kept in tap water for 5 min. In the same way, the antioxidants enzymes, such as SOD and CAT, and cardiac biomarkers, such as cTn-T were also assessed in the cardiac tissues of all experimental groups. A microscope (BX51; Olympus, Tokyo, Japan) was used to capture the images of the sections.

### 4.11. Transmission Electron Microscopy

Heart tissue excised from the rats was perfused with 0.1 M sodium phosphate buffer (pH 7.4) solution containing 2.5% glutaraldehyde and 2% formalin for 15 min at 48 °C according to the protocol described by Watanabe and Yamada [54]. Then the tissue was kept for 2 h at 48 °C in 1% osmium tetroxide contained buffer solution for post fixation. After dehydration, the heart tissue with a various concentration series of alcohol (70–100%), embedded in spurr1 resin. Reichert Ultracut 1 microtome was used for cut the heart tissue with 90 nm thickness. The grids were counterstained with 4% uranyl acetate and 0.4% lead citrate solutions and examined in a Jeol JEM 1010 transmission electron microscope (Peabody, MA, USA) at 80 kV. This assay was performed at the Centre for Cellular and Molecular Biology, Hyderabad, India.

### 4.12. Statistical Analysis

The experimental data was expressed as means ± standard deviation (SD). One-way analysis of variance (ANOVA) was conducted followed by Duncan’s test using SPSS version 19.0 software (Armonk, New York, NY, USA). Statistical significance was considered at *p* < 0.05. Graphs were prepared using GraphPad Prism (Version 5.00, GraphPad Software Inc., San Diego, CA, USA).

## 5. Conclusions

For the first time, our findings demonstrated the potential cardioprotective properties of 6-gingerol against alcohol-induced oxidative stress, apoptosis and architectural damage. The underlying phenomenon may be due to an effective suppression of alcohol-induced ROS protection and/or antioxidant efficacy of gingerol. Further evidence showed that gingerol cotreatment suppressed vital biomarkers of apoptosis and cardiac injury followed by recovered tissue architectural damages. Our findings suggest that 6-gingerol could be a potential therapeutic molecule to use in the treatment of alcohol-induced myocardial injury.

## Figures and Tables

**Figure 1 molecules-27-08606-f001:**
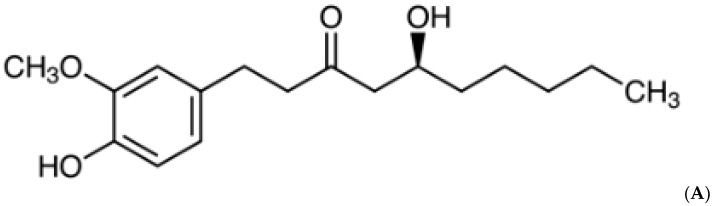
(**A**) Chemical structure of 6-gingerol. (**B**) Changes in intracellular ROS production in control, 6-gingerol, ethanol and 6-gingerol plus alcohol-fed rats.

**Figure 2 molecules-27-08606-f002:**
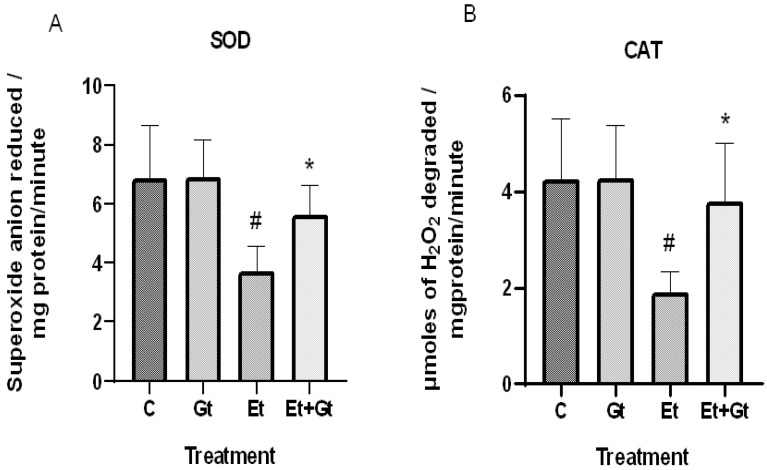
Changes in superoxide dismutase (**A**) and catalase (**B**) activities in cardiac tissue of control (C), 6-gingerol (Gt)-, ethanol (Et)- and ethanol + 6-gingerol (Et + Gt)-treated rats. The data were expressed as mean ± standard deviation (SD). ^#^
*p* < 0.05 compared with control and * *p* < 0.05 compared with ethanol treated groups.

**Figure 3 molecules-27-08606-f003:**
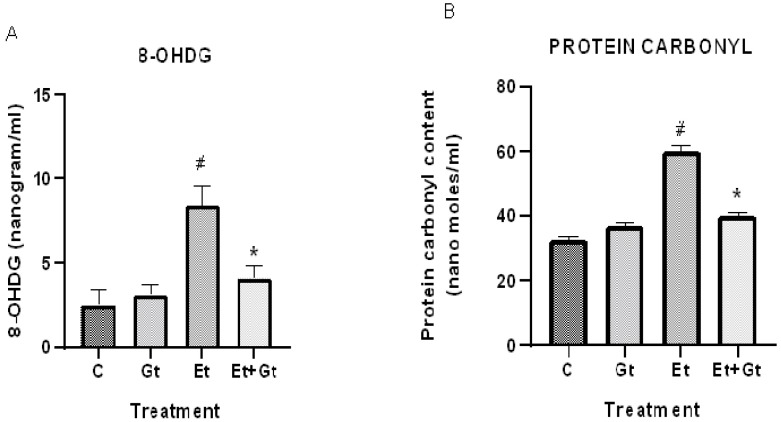
Gingerol treatment suppressed 8-OHDG (**A**) and protein carbonyls (**B**). Assessments were done in control (C), 6-gingerol (Gt)-, ethanol (Et)- and ethanol plus 6-gingerol (Et + Gt)-treated rats. The data were expressed as mean ± SD. ^#^
*p* < 0.05 compared with control and * *p* < 0.05 compared with ethanol-treated groups.

**Figure 4 molecules-27-08606-f004:**
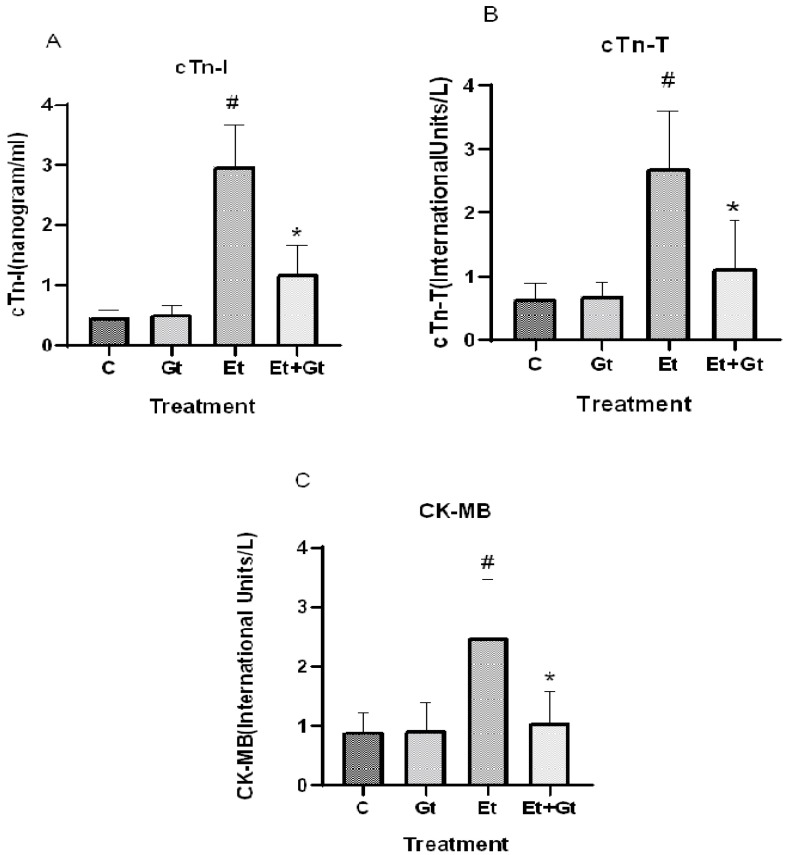
Changes in cTn-I (**A**), cTn-T (**B**) and CK-MB (**C**) levels in plasma of control (C), 6-gingerol (Gt)-, ethanol (Et)- and ethanol plus 6-gingerol (Et + Gt)-treated groups. The data are expressed as mean ± SD. ^#^
*p* < 0.05 compared with control and * *p* < 0.05 compared with ethanol-treated groups.

**Figure 5 molecules-27-08606-f005:**
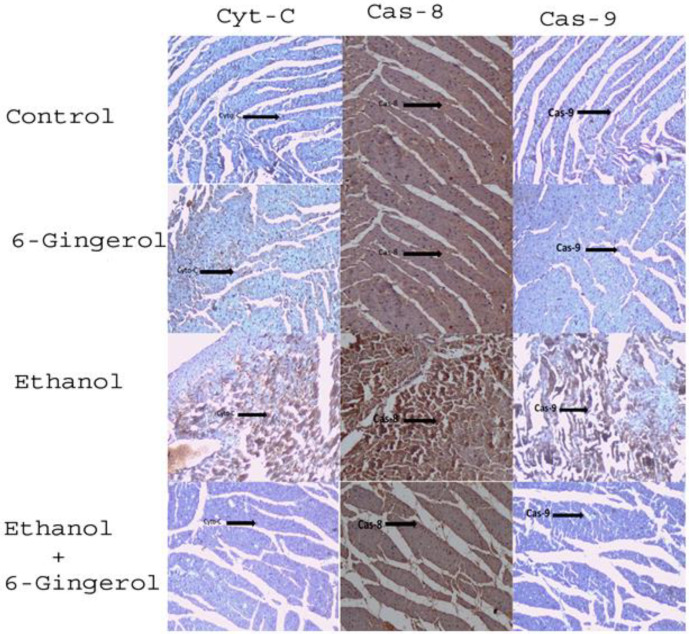
Six-gingerol regulates ethanol induced ROS-mediated apoptosis in cardiac tissue. Expressions of mitochondrial cytochrome C, caspase-8 and caspase-9 in cardiac tissues of control, 6-gingerol, ethanol and ethanol plus 6-gingerol groups. The arrow in each panel indicates the expressions of cytochrome C, caspase-8 and caspase-9 in respective groups.

**Figure 6 molecules-27-08606-f006:**
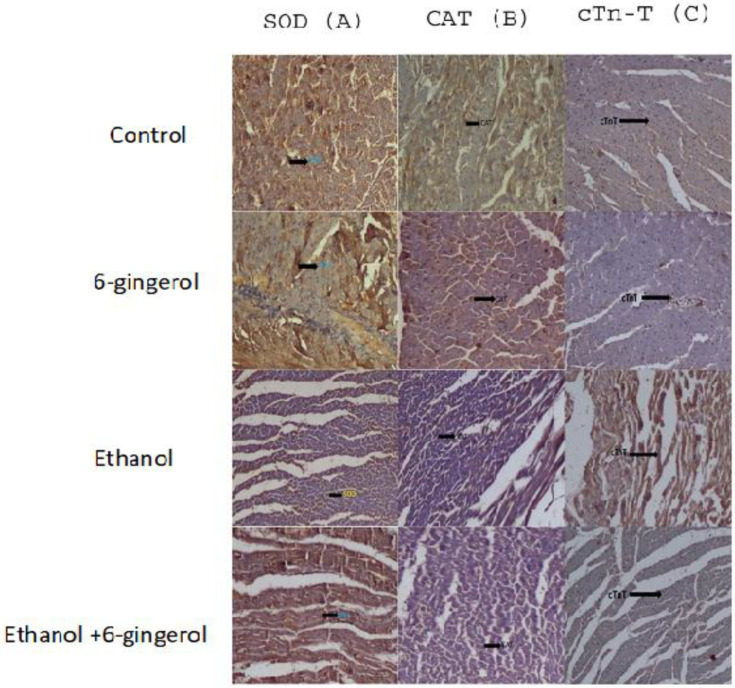
Immunohistochemistry images of SOD (**A**), CAT (**B**) and cTn-T (**C**) in all experimental groups. The arrow in each panel indicates the expressions of SOD, CAT and cTn-T in respective groups.

**Figure 7 molecules-27-08606-f007:**
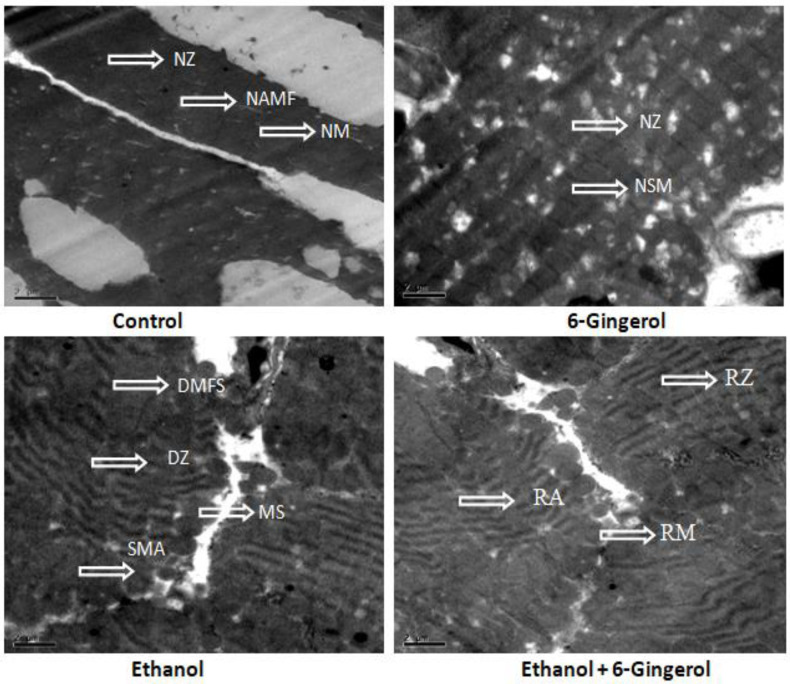
Transmission electron microscopy images of cardiac tissue display alcohol-induced tissue damage and 6-gingerol mediated recovery. The arrow indicates normal structure of Z line (NZ), normal architecture of myofibril (NAMF), normal mitochondria (NM) and normal sarcomere (NSM) in control or 6-gingerol groups. The arrow indicates, degeneration of myofibril structure (DMFS), degeneration of Z lines (DZ), sarcomere alteration (SMA) and mitochondria swelling (MS) in ethanol group. The arrows in ethanol plus 6-gingerol group indicate regeneration of Z line (RZ), regeneration of mitochondria (RM) and reappearance of sarcomere (RA).

**Table 1 molecules-27-08606-t001:** Alcohol-induced dyslipidemia reversed by 6-gingerol. Data were expressed as mean ± SD (N = 6). ^#^
*p* < 0.005 compared with control * *p* < 0.05 compared with ethanol groups. Values were shown in control (C), 6-gingerol (Gt)-, ethanol (Et)- and ethanol plus 6-gingerol (Et + Gt)-treated groups.

Lipid Profile (mg/dL)		Groups		
	C	Gt	Et	Et + Gt
TG	38.83 ± 5.05	39.38 ± 4.17	89.20 ± 6.08 ^#^	44.17 ± 4.98 *
TC	66.94 ± 4.12	64.24 ± 5.13	119.86 ± 5.45 ^#^	72.45 ± 6.13 *
HDL	25.13 ± 2.36	24.51 ± 2.08	12.62 ± 3.06 ^#^	21.33 ± 1.94 *
LDL	40.48 ± 2.11	38.73 ± 1.91	109.11 ± 2.37 ^#^	56.77 ± 1.98 *

## Data Availability

The data which were submitted to this journal is available with corresponding author and first author.

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
