# Peer review of "Cardioprotective Effects of 6-Gingerol against Alcohol-Induced ROS-Mediated Tissue Injury and Apoptosis in Rats"

_molecules, 2022, doi:10.3390/molecules27238606_

Round 1

Reviewer 1 Report

In this manuscript, Ganjikunta et al., studied the cardioprotective effects of 6-gingerol against alcohol-induced ROS-mediated tissue injury and apoptosis in rats. The work is interesting, however, the manuscript is not well written.

Do you know if 6-gingerol affects the expression of two key enzyme in alcohol metabolism, i.e. alcohol dehydrogenase (ADH) and aldehyde dehydrogenase (ALDH)?

What’s the in vivo metabolic pathways of 6-gingerol?

Figure 2. A & B. Changes in superoxide dismutase (A) and catalase (B) activities in cardiac tissue. Have you measured SOD and CAT activity in liver samples?

Figure 4. spelling error. 6-ingerol.

6-gingerol was given at 10 mg/kg, body weight. What’s the rational to use this dose? Is it feasible to human?

Under 4.3. ethanol (EtOH) (received 20% ethanol (v/v) orally) - this is not clear. You need to give the total volumes as well or give total amount in grams /kg BW. Do you give the treatment once a day?

Author Response

Reviewers 1

On behalf of authors, I am highly thankful to the Reviewer for the critical evaluation and valuable comments on our manuscript. We do agree with all the comments that helped us to revise our manuscript quality. Herewith we are providing our point-by-point response to each comment, and all corrections in the manuscript were marked in red for the smooth follow-up.  We believe that the revised version of the manuscript is more suitable for your final decision.

Comment 1: Do you know if 6-gingerol affects the expression of two key enzyme in alcohol metabolism, i.e. alcohol dehydrogenase (ADH) and aldehyde dehydrogenase (ALDH)?

Response:  Authors are thankful to the Reviewer for this in-depth comment on the influence of 6-gingerol on alcohol metabolizing enzymes. In fact we are curious about this for a longtime, but we haven’t done any experiments to demonstrate the 6-gingerol effect of ADH and ALDH. 

                One in vitro study reported that ginger is able to increase ADH activity but decreases ALDH activity. This study urged the need of further in vivo evidence to establish the ginger effects on ADH and ALDH activities (Srinivasan et al. 2019). We assume that ginger or gingerol could influence one of the pathways of alcohol metabolism due to its antioxidant property. It is well documented that 6-gingerol can increase tissue catalase activity in various experimental animal models (Han et al. 2020; Han et al. 2022; Singh et al. 2009). Owing to the fact that catalase metabolizes the alcohol to acetaldehyde (Cederbaum 2012; Zakhari 2006), 6-gingerol-induced increased catalase activity perhaps involve in alcohol metabolism. We further assume this phenomenon may be depends on the degree of catalase increase and/or gingerol concentration. In our study, we also found increased catalase activity and subsequent fewer occurrences of alcohol-induced toxic episodes in gingerol treated group, which indicates alcohol may be rapidly metabolized in the body.

                On the other hand, some studies used food based formulations in combination with ginger, and reported decreased alcohol hangover effects in humans (Sreeraj Gopi et al. 2014; Takahashi et al. 2010). Such beneficial effects of ginger based foods on hangover might be associated with altering the activities of ADH and/or ALDH enzymes. Nevertheless, further studies are necessary to explore the effect of 6-gingerol on ADH and ALDH activities.

                Now we have appropriately discussed this issue in the Discussion of revised manuscript, Page 10, 2nd paragraph.

Comment 2: What’s the in vivo metabolic pathways of 6-gingerol?

Response: We are thankful for this comment. A study by Nakazawa and Ohsawa showed that oral administration of 6-gingerol to rats undergoes conjugation, and omega-1 oxidation and beta oxidation of a phenolic side chain. After oral administration of 6-gingerol to rats, metabolite namely, (S)-[6]-gingerol-4’-O-β-glucuronide (1) was isolated from the bile, and other metabolites, including vanillic acid (2), ferulic acid (3), (S)-(+)-4-hydroxy-6-oxo-8-(4-hydroxy-3-methoxyphenyl)octanoic acid (4), 4-(4-hydroxy-3-methoxyphenyl) butanoic acid (5), 9-hydroxy [6]-gingerol (6), and (S)-(+)-[6]-gingerol (7) were isolated from the β-glucuronidase treated urine samples (Nakazawa and Ohsawa 2002). Quantification of these metabolites indicated that the excretions of 1–6 continued up to 36–48 h, and approximately 48% of the dose was excreted in the bile as glucuronide (1). The total cumulative amounts of 2–7 into the urine were approximately 16% of the dose. Some metabolites of 6-gingerol (1-6) stay about 12 h in the body after oral administration, which means these metabolites may involve in the therapeutic effects of ginger or gingerol. (Nakazawa and Ohsawa 2002).

                Now we have included  the suitable sentence to the Discussion part, Page 10.

Comment 3: Figure 2. A & B. Changes in superoxide dismutase (A) and catalase (B) activities in cardiac tissue. Have you measured SOD and CAT activity in liver samples?

Response: Authors are thankful to the Reviewer for this comment. We would like to bring to Reviewer’s kind notice that we haven’t measured liver SOD or CAT activities in this particular study, as we intended to emphasize the cardioprotective effects of gingerol.

                However, in our previous alcohol-based studies, we found alcohol caused a significant decrease of liver antioxidant enzyme activities, and these enzymes were restored by ginger (Mallikarjuna et al. 2008) and exercise interventions (Mallikarjuna et al. 2010).

Comment 4: Figure 4. spelling error. 6-ingerol.

Response: We are thankful to the Reviewer for letting us to correct our typos. As suggested, the spelling of 6-gingerol was corrected in Figure 4.

Comment 5: 6-gingerol was given at 10 mg/kg, body weight. What’s the rational to use this dose? Is it feasible to human?

Response: We appreciate the Reviewer for this meaningful comment. In our lab, we are continuously doing our studies on ginger and ginger ingredients from the long time. For the gingerol dose, we have done preliminary dose-dependent studies on rats, and identified 10 mg/kg bodyweight as the effective dose without any adverse effects. Besides, other previous animal studies used a range of 6-gingerol dose from 6 to 200 mg/kg bodyweight, and reported such dose of gingerol to be safe to rodents, and pharmacologically effective under various experimental stress conditions (El-Bakly et al. 2012; Han et al. 2022; Hashem et al. 2017; Xu et al. 2018; Young et al. 2005). Based on these previous reports, we therefore selected gingerol dose (10 mg/kg bodyweight) in our study.

                Yes, the given dose of 6-gingerol for rat model is feasible for humans. For instance, a human dose-escalation study on adults reported no serious adverse effects with an oral administration of 6-gingerol from low concentration (100 mg) to high concentration 2000 mg (Zick et al. 2008). According to the human equivalent dose (HED) conversion formula, the dose we used in our rat model (10 mg/kg) is equals to 1.621 mg/kg for adult, and it would be around 97 mg per adult with an average bodyweight of 60 kg. This dose of 6-gingerol is still less than the reported safety dose, and would be feasible for humans.

                We consider this information is important and therefore, suitable explanation was included in our revised manuscript, Discussion part first paragraph (Page 7).

Comment 6: Under 4.3. ethanol (EtOH) (received 20% ethanol (v/v) orally) - this is not clear. You need to give the total volumes as well or give total amount in grams /kg BW. Do you give the treatment once a day?

Response: We are apologizing for not providing the weight of the alcohol. As suggested, now we provided the grams of ethanol per kg bodyweight in our revised manuscript, Page 11. We further confirm that we gave the treatment once a day, and this information was mentioned in the revised version, Page 11.

Bibliography:

Cederbaum AI (2012) Alcohol Metabolism. Clinics in Liver Disease 16(4):667-685 doi:https://doi.org/10.1016/j.cld.2012.08.002

El-Bakly WM, Louka ML, El-Halawany AM, Schaalan MF (2012) 6-gingerol ameliorated doxorubicin-induced cardiotoxicity: role of nuclear factor kappa B and protein glycation. Cancer chemotherapy and pharmacology 70(6):833-841

Han X, Liu P, Liu M, et al. (2020) [6]-Gingerol ameliorates ISO-induced myocardial fibrosis by reducing oxidative stress, inflammation, and apoptosis through inhibition of TLR4/MAPKs/NF-κB pathway. Molecular Nutrition & Food Research 64(13):2000003

Han X, Yang Y, Zhang M, et al. (2022) Protective Effects of 6-Gingerol on Cardiotoxicity Induced by Arsenic Trioxide Through AMPK/SIRT1/PGC-1α Signaling Pathway. Frontiers in Pharmacology 13 doi:10.3389/fphar.2022.868393

Hashem RM, Rashed LA, Hassanin KMA, Hetta MH, Ahmed AO (2017) Effect of 6-gingerol on AMPK- NF-κB axis in high fat diet fed rats. Biomedicine & Pharmacotherapy 88:293-301 doi:https://doi.org/10.1016/j.biopha.2017.01.035

Mallikarjuna K, Chetan PS, Reddy KS, Rajendra W (2008) Ethanol toxicity: Rehabilitation of hepatic antioxidant defense system with dietary ginger. Fitoterapia 79(3):174-178

Mallikarjuna K, Shanmugam KR, Nishanth K, et al. (2010) Alcohol-induced deterioration in primary antioxidant and glutathione family enzymes reversed by exercise training in the liver of old rats. Alcohol 44(6):523-529 doi:https://doi.org/10.1016/j.alcohol.2010.07.004

Nakazawa T, Ohsawa K (2002) Metabolism of [6]-gingerol in rats. Life Sciences 70(18):2165-2175 doi:https://doi.org/10.1016/S0024-3205(01)01551-X

Singh AB, Singh N, Maurya R, Kumar A Anti-hyperglycaemic, lipid lowering and anti-oxidant properties of (6)-gingerol in db/db mice. In, 2009.

Sreeraj Gopi RG, Thankachen RU, Sriraam V, Abirami S (2014) Studies on the effectiveness and safety of anti hangover drink (Oh! K) in reducing cocktail (alcohol) induced hangover symptoms in adult male social drinkers. 

Srinivasan S, Dubey KK, Singhal RS (2019) Influence of food commodities on hangover based on alcohol dehydrogenase and aldehyde dehydrogenase activities. Current Research in Food Science 1:8-16 doi:https://doi.org/10.1016/j.crfs.2019.09.001

Takahashi M, Li W, Koike K, Sadamoto K (2010) Clinical effectiveness of KSS formula, a traditional folk remedy for alcohol hangover symptoms. Journal of Natural Medicines 64(4):487-491 doi:10.1007/s11418-010-0430-9

Xu T, Qin G, Jiang W, Zhao Y, Xu Y, Lv X (2018) 6-Gingerol Protects Heart by Suppressing Myocardial Ischemia/Reperfusion Induced Inflammation via the PI3K/Akt-Dependent Mechanism in Rats. Evidence-Based Complementary and Alternative Medicine 2018:6209679 doi:10.1155/2018/6209679

Young H-Y, Luo Y-L, Cheng H-Y, Hsieh W-C, Liao J-C, Peng W-H (2005) Analgesic and anti-inflammatory activities of [6]-gingerol. Journal of Ethnopharmacology 96(1):207-210 doi:https://doi.org/10.1016/j.jep.2004.09.009

Zakhari S (2006) Overview: how is alcohol metabolized by the body? Alcohol research & health 29(4):245

Zick SM, Djuric Z, Ruffin MT, et al. (2008) Pharmacokinetics of 6-Gingerol, 8-Gingerol, 10-Gingerol, and 6-Shogaol and Conjugate Metabolites in Healthy Human Subjects. Cancer Epidemiology, Biomarkers & Prevention 17(8):1930-1936 doi:10.1158/1055-9965.epi-07-2934

Reviewer 2 Report

The authors evaluate the cardioprotective action of 6-gingerol in male rats receiving ethanol.

The authors evaluate the action of reactive oxygen species on SOD and CAT enzymes. In addition, they evaluate some blood parameters.

The manuscript has potential, however some fundamental methodological criticalities should be clarified:

1) What is the administration route of ethanol?

2) What is the route of administration of 6-gingerol?

3) It is unclear whether the authors induced alcoholic fatty liver disease before evaluating the effects of 6-gingerol.

4) Do the authors want to give the 6-gingerol treatment a preventive or therapeutic value?

5) Figure 1b is unclear.

Author Response

Reviewers 2

On behalf of authors, I am highly thankful to the Reviewer for the critical evaluation and valuable comments on our manuscript. We do agree with all the comments that helped us to improve our manuscript quality. Herewith we are providing our point-by-point response to each comment, and all corrections in the manuscript were marked in red for smooth follow-up.  We believe that the revised version of the manuscript is more suitable for your final decision.

Comment 1: What is the administration route of ethanol?

Response:  Authors are thankful to the Reviewer for this meaningful comment. In our study, ethanol was orally administered with the gavage. Now this information of clearly describe in the revised manuscript, Page 11.

Comment 2: What is the route of administration of 6-gingerol?

Response: We are apologizing for not giving the administration details of 6-gingerol in our previous version of the manuscript. Six-gingerol was orally administered rats with the gavage. As suggested, now we have included this information in the revised manuscript, Page 11.

Comment 3: It is unclear whether the authors induced alcoholic fatty liver disease before evaluating the effects of 6-gingerol.

Response: We appreciate Reviewer for this in-depth view point. We would like to bring to Reviewer’s kind notice that in this particular study, we haven’t measured the alcoholic fatty liver disease, as we primarily focused on myocardial injury by alcohol and recovery by 6-gingerol. However, our previous studies demonstrated that alcohol intoxication caused a significant decrease of hepatic antioxidant enzyme activities and peroxidation of lipids in young and old rats (Mallikarjuna et al. 2008; Mallikarjuna et al. 2010). 

Comment 4: Do the authors want to give the 6-gingerol treatment a preventive or therapeutic value?

Response: Authors are thankful to the Reviewer for this interesting comment. We could have mentioned this in our manuscript clearly. In this study design, we demonstrated the therapeutic effects of 6-gingerol by co-administrating with alcohol. The treatment for all experimental groups was started on the same and ended on the same day as well. The alcohol-induced adverse effects and gingerol protective effects were compared between the groups. In this case, the protective effects of 6-gingerol in alcohol co-treated group (Gt+Et group) were compared with alcohol alone treated group, which implies therapeutic and preventive effects. We believe that 6-gingerol could also be a potent molecule to prevent the alcohol-induced myocardial injury. The relevant changes were made in our revised manuscript wherever applicable.

Comment 5: Figure 1b is unclear.

Response: We are thankful to the Reviewer for letting us to improve the Figure quality. As suggested, we have improved the Figure 1B quality. If necessary, the original image will be sent to relevant staff during production time.

Bibliography:

Mallikarjuna K, Chetan PS, Reddy KS, Rajendra W (2008) Ethanol toxicity: Rehabilitation of hepatic antioxidant defense system with dietary ginger. Fitoterapia 79(3):174-178

Mallikarjuna K, Shanmugam KR, Nishanth K, et al. (2010) Alcohol-induced deterioration in primary antioxidant and glutathione family enzymes reversed by exercise training in the liver of old rats. Alcohol 44(6):523-529 doi:https://doi.org/10.1016/j.alcohol.2010.07.004

Round 2

Reviewer 1 Report

I recommend to accept this revised version for publication in Molecules.